# Calibrating Laser Three-Dimensional Projection Systems Using Binocular Vision

**DOI:** 10.3390/s23041941

**Published:** 2023-02-09

**Authors:** Dabao Lao, Yukai Wang, Fang Wang, Chao Gao

**Affiliations:** 1College of Automation, University of Science and Technology Beijing, Beijing 100083, China; 2Institute of Microelectronics of the Chinese Academy of Sciences, Beijing 100029, China

**Keywords:** binocular vision, galvo scanner, laser projection, calibration

## Abstract

A laser three-dimensional (3D) projection system is an auxiliary system in intelligent manufacturing. It works with a positioning system in practical applications. This study proposes a calibration method for laser 3D projection systems based on binocular vision. The significance of the binocular vision positioning function for the calibration process was analyzed. Two calibration methods for laser 3D projection systems based on the binocular vision positioning function were proposed. One method involves simplified calculation models and another used data to solve the conversion relationship. The experimental calibration of the projection system was performed using data to directly solve the conversion relationship. The experiment demonstrated the simplicity of the proposed calibration method. The calculation time was less under the 3D laser projection system based on binocular vision. Moreover, the mean calibration error was 0.38 mm at a working distance of 1.8–2.2 m.

## 1. Introduction

Laser three-dimensional (3D) projection is one of the innovative uses of lasers, compensating for the lack of manual processing. Laser 3D projection is an auxiliary tool for precise large-scale equipment production. It uses the vision theory to produce the desired pattern or outline by directing the laser beam at the exact location of the projected target through a fast-moving deflection mechanism [1]. Laser 3D projection is increasingly used in manufacturing processes, such as composite laying, pattern spraying, and assembly guidance, and the part welding of large-scale components with complex shapes in aerospace, ship building, automobile manufacturing, and other large-scale equipment manufacturing industries [2]. It realizes intelligent production while minimizing errors and increasing productivity by functioning as an auxiliary measurement and marking system [3].

The laser deflection device is the primary part of the laser 3D projection system. The operating speed and accuracy of the laser 3D projection system depend on its deflection speed and accuracy. Compared with other deflection devices, the galvo scanner has significant benefits in precision and speed owing to the galvanometer it uses for deflection [4]. The deflection device of a laser 3D projection system is typically a galvo scanner. To determine the relationship between the deflection values of the galvo scanner and target coordinates, most laser 3D projection systems require manual adjustment of the galvo scanner’s deflection and project the laser to a known coordinate point on the projected target. This process is the calibration of the laser 3D projection system [5].

Calibration is essential before operating the 3D laser projection system. Two typical methods are used for calibrating the system: The first method evaluates the physical model of the galvo scanner, establishes the coordinate system of the galvo scanner, and carries out the conversion of the coordinate systems. The other method uses a large amount of data to fit the nonrelational relationship between the input and output. A method for projecting graphic outlines with a laser projector was created by Rueb K D in 1997 [6]. This method manually calibrates the laser projector to match a recognized object coordinate system. This method is labor-intensive, and the production process of the galvo scanner influences the calibration accuracy. Some researchers have looked into simpler algorithms and auxiliary calibration tools to streamline the calibration procedure and enhance the usability of laser 3D projection systems. A laser projector that projects a 3D image onto an object was invented by Kaufman et al. [7]. The projector measures the distance between the instrument and projected target using a time module connected to an optical module, which adds this parameter to the calibration procedure. Cui et al. [8] applied the imaging principle of a camera to laser projection and used the principle of the camera imaging model to calibrate the projection system. Guo Lili [9] proposed using a laser reflection device to acquire the galvo scanner’s exit coordinates in 3D space, shortening the projection time and steps. These techniques simplify the calibration procedure to some extent. However, they do not address the issue of repeating the calibration when the position of the target changes. In 2011, Rueb et al. [10] suggested using a laser template to deduce the projection target’s location. Kaufman et al. [11] also developed a laser 3D projection motion tracking component which enables the system to update the pose according to the target. These techniques address the issue of repeated calibration; however, the overall structural layout of the system becomes more complex. Tobais et al. [12] built a laser 3D projection system under a single camera using Gaussian fitting (GP), ridge regression, artificial neural network (ANN), support vector regression (SVR), and other fitting algorithms to study the relationship between the driving values of the galvo scanner and image coordinates. This calibration method does not consider the complex physical model and is not affected by the system structure design. This calibration method has been widely studied in recent years.

The visual measurement method has the advantages of convenient operation and simple positioning and is suitable for calibrating laser 3D projection systems. A corrected laser 3D scanning system may comprise one or more cameras set in a fixed orientation on the scanner for system calibration, as suggested by Morden et al. [13]. Based on this, Qi et al. [14] proposed the stereo vision laser galvanometric scanning system and a system calibration method using plastic thin film targets, and applied it to cutting duck feathers for badminton shuttle manufacture. Its use scenario is relatively simple, and it needs to be recalibrated when changing the position. Tu et al. [15] proposed a calibration method that builds a neural network and takes the digital control signal at the drives of the GLS system as input and the space vector of the corresponding outgoing laser beam as output. This paper makes some improvements to the calibration method of the neural network. This method takes the set of coordinates in the target point cloud as input and the deflection values of the galvo scanner as output and supports the dynamic adjustment of the input and output data. The accurate calibration of the 3D laser projection system is achieved through this method combined with the pose relationship between the camera and galvo scanner. This method is simpler to use, less repetitive than the conventional approach, more precise and intelligent, and adaptable to various application scenarios.

## 2. Calibration Method Based on Binocular Vision

### 2.1. Positioning Theory for Binocular Vision

The binocular vision positioning theory is based on the geometric model of similar triangles in the binocular field of view. A mathematical model can be used to explain how the camera captures an image of the real object. The model comprises three coordinate systems: image, camera, and world. The two-dimensional (2D) image coordinate system has a plane identical to the physical imaging plane of the camera. The pixel layout determines the directions of the X and Y coordinate axes, with the unit of measurement in millimeters. The center of the physical imaging plane serves as the origin of the image coordinate system. The origin of the camera coordinate system is the camera optical center O. The XC and YC coordinate axes are parallel to the image coordinate system. The ZC coordinate axis coincides with the optical axis of the camera and is perpendicular to the physical imaging plane. The experimenter typically establishes the world coordinate system. The location of the origin and axis of the coordinates can be selected randomly; typically, it is placed on the measured object to facilitate description. Figure 1 shows the positional relationship of the three coordinate systems.

Assume a point (*x_c_*,*y_c_*,*z_c_*) on the camera coordinate system. This point corresponds to a point (*x′*,*y′*) on the image coordinate system after the camera captures the image. The coordinate transformation relationship from the image coordinate system to the camera coordinate system can be expressed as Equation (1) [16]:(1)$$Zc\left[\begin{array}{c} x^{\prime }\\ y^{\prime }\\ 1 \end{array}\right]=\left[\begin{array}{cccc} f & 0 & 0 & 0\\ 0 & f & 0 & 0\\ 0 & 0 & 1 & 0 \end{array}\right]\left[\begin{array}{c} x_{c}\\ y_{c}\\ z_{c}\\ 1 \end{array}\right],$$
where *f* is the focal length of the camera, determined during camera calibration, and *Zc* is a coefficient.

The conversion relationship between the coordinates (*x_c_*,*y_c_*,*z_c_*) of the camera coordinate system and coordinates (*x_n_*,*y_n_*,*z_n_*) of the world coordinate system can be expressed as Equation (2) [16]:(2)$$\left[\begin{array}{c} \begin{array}{c} x_{c}\\ y_{c} \end{array}\\ \begin{array}{c} z_{c}\\ 1 \end{array} \end{array}\right]=\left[\begin{array}{cc} R & t\\ 0^{T} & 1 \end{array}\right]\left[\begin{array}{c} \begin{array}{c} x_{n}\\ y_{n} \end{array}\\ \begin{array}{c} z_{n}\\ 1 \end{array} \end{array}\right].$$

Equations (1) and (2) can be used to obtain the camera imaging model, where *R* and *t* are the rotation matrix and translation vector of the coordinate system transformation, respectively, and *a_x_*, *a_y_*, *u*_0_, and *v*_0_ are the internal parameters of the camera. The imaging model of the camera can be expressed as Equation (3) [16]:(3)$$Zc\left[\begin{array}{c} x^{\prime }\\ y^{\prime }\\ 1 \end{array}\right]=\left[\begin{array}{cccc} a_{x} & 0 & u_{0} & 0\\ 0 & a_{y} & v_{0} & 0\\ 0 & 0 & 1 & 0 \end{array}\right]\left[\begin{array}{cc} R & t\\ 0^{T} & 1 \end{array}\right]\left[\begin{array}{c} \begin{array}{c} x_{n}\\ y_{n} \end{array}\\ \begin{array}{c} z_{n}\\ 1 \end{array} \end{array}\right].$$

When two cameras are used by the binocular vision system to capture photographs of the same target from different angles, the difference between the two images represents the information on the object’s three dimensions. Disparity can be calculated when the two image planes are completely coplanar, and the lines are aligned. This is known as the binocular stereoscopic correction process, dealing with the perspective transformation of the image. After the binocular stereoscopic correction, the three-dimensional coordinates of the space can be solved using the physical model shown in Figure 2: P is a point on the object, Z is the distance from point P to the focal plane, and *f* is the focal length. *P_l_* and *P_r_* are the projection points of point P on the focal planes of the left and right cameras with abscissas *X_l_* and *X_r_*, respectively, and T is the distance between the optical centers of the two cameras.

Based on the geometric relationship of similar triangles, the spatial coordinates (*x/w*, *y/w*, *z/w*) of point P in the camera coordinate system can be expressed as Equation (4) [17].
(4)$$\left[\begin{array}{c} \begin{array}{c} x\\ y \end{array}\\ \begin{array}{c} z\\ w \end{array} \end{array}\right]=\left[\begin{array}{cccc} 1 & 0 & 0 & -su_{0}\\ 0 & 1 & 0 & -sv_{0}\\ 0 & 0 & 0 & f\\ 0 & 0 & -\frac{1}{T} & 1 \end{array}\right]\left[\begin{array}{c} \begin{array}{c} X_{r}\\ Y_{r} \end{array}\\ \begin{array}{c} d\\ 1 \end{array} \end{array}\right],$$
where (*X_r_*,*Y_r_*) are the projected point coordinates of P in the stereo-corrected image, *s* is the millimeter size of the pixel, *d* = *X_l_* − *X_r_* is the disparity between the coordinates of the two cameras, and *w* is the scale factor.

Any known coordinate point in the world coordinate system can be converted to the camera coordinate system using the aforementioned method. The unique conversion relationship between the world and camera coordinate systems can be solved if the four sets of matching coordinates in each of the two coordinate systems are known. The corresponding relationship between the deflection values of the galvo scanner and coordinates of the projected target is based on this conversion relationship.

### 2.2. Calibration Using Binocular Vision

The relationship between the camera and the projected target can be determined using the positioning function of binocular vision in Section 2.1. The pose relationship between the camera and galvo scanner should be solved to complete the laser 3D projection system calibration. Two methods are studied to solve the relationship between the camera coordinates and deflection values of the galvo scanner.

#### 2.2.1. Simplifying the Computational Model of Galvanometers Using Binocular Vision

The galvo scanner structure can be simplified into a physical model shown in Figure 3. The digital circuit can convert the digital signals *d_x_* and *d_y_* into voltage signals *V_x_* and *V_y_* to drive the motor to deflect a horizontal angle *θ_x_* and pitch angle *θ_y_*. The mirror fixed on the motor also deflects the same angle as a galvanometer; a galvo scanner coordinate system with the center of mirror 2 as the origin, the coordinate axis *X_g_* parallel to the rotation axis of motor 2, and the coordinate axis *Z_g_* parallel to the rotation axis of motor 1. After the laser is emitted from the transmitter, it can theoretically be projected to any position (*x*,*y*,*z*) in the front space after being deflected twice. The relationship between the deflection value of the galvanometer and coordinate can be expressed as Equation (5):(5)$$\{\begin{array}{l} z=dp\\ y=dp\times \tan2\theta_{y}\\ x=e\times \tan2\theta_{x}+dp\times \tan2\theta_{x}/\cos2\theta_{y} \end{array},$$
where *e* is the distance between the central axes of the two mirrors, determined during galvo scanner installation, and *dp* is the *Z*-axis coordinate in the galvo scanner coordinate system, typically obtained by the laser exit time *tc* or measured by a laser reflection device.

Using a binocular camera simplifies solving for the coordinate value *z*. When installing the hardware platform of the laser 3D projection system, the binocular camera is installed parallel to the *X_g_OZ_g_* plane of the galvo scanner coordinate system and relatively fixed. The same applies for the plane to be projected. The deflection values of the two galvanometers are adjusted to zero. The current output coordinates are (0,0,*z*) in the galvo scanner coordinate system. The galvo scanner is adjusted to deflect an angle *θ_y_* arbitrarily. The *Z_g_* axis coordinate does not change because the projection plane is placed parallel to the *X_g_OY_g_* plane. The exit coordinate in the galvo scanner coordinate system is (0,*y*,*z*). As shown in Figure 4, the positioning function of binocular vision can be used to solve coordinate *z* directly when the binocular camera is installed parallel to the *X_g_OZ_g_* plane. The coordinate value y in the galvanometer coordinate system equals the coordinate value in the camera coordinate system. In this case, *z = y*/tan2*θ_y_*.

After determining the coordinate value *z* and deflection angles *θ_x_* and *θ_y_* of the galvo scanner, the precise coordinates (*x*,*y*,*z*) on the projected target can be obtained using Equation (5) by controlling the deflection of the galvo scanner arbitrarily. This method is used for obtaining four groups of coordinates of the galvo scanner and camera coordinate systems to determine the relationship between them and solve the affine transformation matrix.

The physical model analysis method of the scanning galvanometer is established under ideal hardware installation conditions. In reality, whether the laser light source is vertically irradiated on the center of the mirror and whether the central axis of the two mirrors is vertical or not significantly influence the calibration result when designing the hardware platform of the 3D projection system. Therefore, the aforedescribed method is only suitable for some specific scenarios.

#### 2.2.2. Solve Calibration Directly Using Data-Driven Approach

In calibrating the system using the binocular vision, the data can be directly used to solve the conversion relationship between the deflection values of galvo scanner and coordinates of the camera coordinate system. Fitting their relationship with least squares is a straightforward method. Given N groups of different deflection values of the galvo scanner,
(6)$$D=\left\{\left(dx_{i},dy_{i}\right)|i=1,2\cdots N\right\},$$
and the corresponding coordinates of camera coordinate system,
(7)$$V=\left\{\left(x_{i},y_{i},z_{i}\right)|i=1,2\cdots N\right\},$$
the least-squares solution of the conversion relationship *H* between the two can be determined using Equation (8):(8)$$H={\left(V^{⊺}V\right)}^{-1}V^{⊺}D.$$

Although the least-squares method is straightforward and simple, it has several limitations because it is susceptible to outlier interference. Moreover, the relationship between the model input and output is nonlinear, whereas the least-squares fitting function is a linear relationship. When the *z* coordinate in the 3D coordinate is fixed, the laser is deflected and projected on a plane by the galvo scanner. The relationship between the deflection values of the galvo scanner and coordinates of the camera coordinate system is expressed as follows:(9)$$\left(x/\tan\theta_{0}-e^{2}\right)-y^{2}=d^{2}.$$

Equation (9) shows that when the galvo scanner is linearly driven, the laser output graph obtained is a closed graph with a hyperbola, rather than an ideal rectangle, as shown in Figure 5. The relationship is also obtained when the galvo scanner works in a 3D space; therefore, using linear techniques, such as the least-squares method, to directly solve it is inappropriate.

A neural network is a suitable direct solution technique with good nonlinear function approximation capabilities. The single hidden layer neural network shown in Figure 6 can approximate the nonlinear relationship accurately [18]. For the calibration model in this study, the input and output dimensions were three and two, respectively. No complex network model was required to solve the single hidden layer neural network model to quickly and accurately determine the conversion between the deflection values of the galvo scanner and coordinates of the camera coordinate system.

In the calibration model, the number of neurons in the input and output layers were three and two, respectively. Assuming that there are *N* groups of different coordinate values as input $v_{i}$, the network output can be expressed as follows [15]:(10)$$f\left(v_{i}\right)=\sum_{j=1}^{L}\beta_{i}G_{j}\left(\omega_{j},b_{j},v_{i}\right),i=1\cdots N,$$
where *L* is the number of neurons in the hidden layer; it is typically set to be equal to or close to the number of samples *N* to obtain a better fitting relationship. *ω* and *b* represent the weights and biases from the input layer to the hidden layer, respectively. *β* represents the weight from the hidden layer to the output layer, and *G* is the activation function.

To simplify the solution, the input–output relationship of the network is simplified as *D = βH*, and *H* is given by Equation (11):(11)$$H=H\left(\omega_{1},\cdots ,\omega_{L},b_{1},\cdots ,b_{L},x_{1},\cdots ,x_{N}\right).$$

When the number of neurons in the hidden layer is sufficiently large (greater than *N*), *ω* and *b* in *H* are randomly selected, where *ω* is a set of random values with a mean of 1, and *b* is a set of random values with a mean of 0. The sample points in the interval can be arbitrarily interpolated; that is, the network can approximate any sample point with zero error. Moreover, the matrix *H* must be invertible; that is, there must be an inverse matrix or a pseudo-inverse matrix. The least-squares solution of *β* can be obtained using Equation (12):(12)$$\beta =H^{+}D,$$
where $H^{+}$ is the generalized inverse of *H*. The known random values of *ω* and *b* and the calculated *β* are used to determine the mapping relationship. This method refers to the extreme learning machine (ELM) algorithm and calculates the least-squares solution by randomly selecting the weights and biases between the input and hidden layers. The smallest empirical risk can be obtained conveniently, and new data can be added at any time. The solution is also the smallest two-norm solution among all least-squares solutions, as shown in Equation (13) [1]:(13)$$\left|\left|\beta_{0}\left|\left|=\right|\right|H^{+}D\right|\right|\leq \left|\left|\beta \right|\right|,\forall \beta \in \left\{\left|\left|H\beta -D\right|\right|\leq \left|\left|H\zeta -D\right|\right|,\forall \zeta \in R^{L\times N}\right\}.$$

This results in a good network generalization performance.

When the camera coordinate system coordinates are given, the deflection values of the galvo scanner can be obtained by solving the network output, and the conversion relationship can be determined through the neural network.

Combining the methods in Section 2.1 and Section 2.2.2, a complete laser 3D projection system calibration process can be completed. The block-level diagram of the calibration is shown in Figure 7.

## 3. Projection System Calibration Experiment and Result Analysis

### 3.1. Projection System Calibration Experiment

The calibration process of the laser 3D projection system comprises the following steps:The relative position of the camera and galvo scanner in the laser 3D projection device remain unchanged. Moreover, a set of drive values is predetermined to deflect the laser by an angle such that it is projected on the projection working range (1.8–2.2 m in this experiment) to generate a spot.The laser 3D projection system is adjusted such that the binocular camera can clearly obtain the laser spot image. Moreover, the internal and external parameters of the binocular camera are calibrated. The positioning function of binocular vision is used to obtain the spatial coordinates of the spot and match the deflection values of the galvo scanner.Steps one and two are repeated to obtain the 3D coordinates of multiple groups of different outgoing laser spots and matching deflection values of the galvo scanner. The neural network is used to solve the relationship between the coordinates of the camera coordinate system and deflection values of the galvanometer.The position of the projected target with marking points is fixed, and a world coordinate system is established. The positioning function of binocular vision is used to obtain the coordinates of the four marking points on the projected target in the camera coordinate system. The transformation matrix between the two coordinate systems is solved after determining the coordinates of the four points.The corresponding relationship between the deflection values of the galvo scanner and coordinates of the coordinate system of the object to be projected can be obtained by combining steps three and four. Thereafter, the system calibration is completed.

The relationship between the camera coordinate system coordinates and deflection values of the galvo scanner is solved once because the positional relationship between the galvo scanner and binocular camera is relatively fixed and unchanged. Only the positioning function of the binocular vision is required to determine the conversion relationship between the camera coordinate system and the coordinate system of the projected target for calibrating the laser 3D projection system. Figure 8 shows the flow chart of the calibration procedure.

A metal plate, shown in Figure 9, was customized to verify the calibration accuracy during the calibration experiment of the laser 3D projection system. The metal plate had four holes, used as the target to be located. The reflection effect is good, and the position of the projected laser point can be measured and collected conveniently. The processing error of the metal plate was less than 0.005 mm using vernier calipers and other measuring tools for multiple measurements, satisfying the requirements. A laser 3D projection system was designed to strictly meet the calibration conditions, as shown in Figure 10a. The system comprised two modules: visual positioning and laser projection. The visual positioning module comprised two cameras. The system used two identical sets of cameras and lenses to ensure that the focal length and other lens parameters, are as consistent as possible. This helped satisfy the requirements of the binocular vision positioning principle discussed in Section 1 and improved the calibration accuracy. The installation position of the binocular camera was relatively fixed with the installation position of the galvo scanner. The binocular camera was accurately calibrated before use, and the influence of distortion on the camera imaging was eliminated. The laser projection module comprised a galvo scanner, a galvo scanner control card, and other optics. The galvo scanner is controlled by the galvo scanner control card connected to the computer. The laser transmitter emits laser light through the collimating beam expander, dynamic focus lens, and galvo scanner. Subsequently, millimeter-level laser beams can be projected on targets at different distances. Figure 10b shows the physical diagram of the system.

A program that projects 55 × 50 laser dots each time onto three metal plates was created to form a dot matrix. The position of the metal plate was freely moved four times within the working range of 1.8–2.2 m, and 11,000 sets of data were collected. The network model with the number of hidden layer neurons *L* = 12,000 and penalty factor *C* = 5000 was used to train the data. Computing was performed using an ASUS laptop with an AMD Ryzen 5 4600H CPU and an NVIDIA GeForce GTX 1660 Ti graphics card. The squeeze function was used as the activation function, according to the discussion and analysis in Section 2.2.2. The weights *ω* and bias *b* were recorded from the input layer to the hidden layer, and the weights *β* were recorded from the hidden layer to the output layer. When the 3D vector *X* = (*X*,*Y*,*Z*) of the coordinates of the camera coordinate system is known, the 2D vector *T* of the drive value of the galvo scanner can be obtained using Equation (14):(14)$$T=\left(tx,ty\right)=\frac{\beta }{e^{-\left(X\cdot \omega +b\right)}+1}.$$

### 3.2. Results Analysis

The calibration findings were verified and evaluated during the laser 3D projection system calibration. A set of drive values of the galvo scanner (*tx*,*ty*) and the coordinates of the camera coordinate system (*X*,*Y*,*Z*) of the laser were projected onto the metal plate under these values as a test set. For example, there is a designed coordinate point (*x_p_*,*y_p_*,*z_p_*) in the point cloud of the shape “3” formed by the CAD digital model file in Figure 11a. The deflection values of the galvo scanner (*txr*,*tyr*) at this coordinate value can be obtained by using the calibration result as Equation (15):
(15)$$\left(txr,tyr\right)=\frac{\beta }{e^{-\left(\left(x_{p},y_{p},z_{p}\right)\cdot \omega +b\right)}+1}.$$

Using the solved (*txr*,*tyr*) to drive the galvo scanner will generate a laser spot on the projection target, as shown in Figure 11b. The positioning function of the binocular camera can solve the coordinates of the spot (*x_r_*,*y_r_*,*z_r_*) in the camera coordinate system. The distance *d* between the 3D coordinates (*x_p_*,*y_p_*,*z_p_*) designed in CAD and the actual projected 3D coordinates (*x_r_*,*y_r_*,*z_r_*) is used as the error of the analysis system, as shown in Figure 11c:(16)$$d=\sqrt{{\left(x_{r}-x_{p}\right)}^{2}+{\left(y_{r}-y_{p}\right)}^{2}+{\left(z_{r}-z_{p}\right)}^{2}}.$$

A total of 1000 designed coordinate points were selected and the above process was repeated to obtain 1000 error data. Figure 12 shows the error with the number of test sets as *Nt* = 1000. The maximum and average deviations of the projection laser spot was 0.87 mm and 0.37 mm, respectively.

Solving the conversion between the coordinates of the camera coordinate system and the deflection values of the galvo scanner uses a single hidden layer neural network method, so the error in establishing the relationship between the scanning galvanometer and camera coordinate systems is mainly related to the solution process of the neural network. When the activation function is selected, the training data volume was presumed to influence the calibration accuracy. To test this hypothesis, different numbers of training sets were selected, and the appropriate number of neurons in the hidden layer was used for system calibration. The same test set was used to analyze the calibration error. Table 1 shows the mean deviations, maximum deviations, and computing time. The more data in the training set, the smaller the deviation and the better the calibration effect, but the longer the computing time.

It can be seen from Table 1 that the more data in the training set, the smaller the impact on the error, but the computing time will increase quadratically, as shown in Figure 13. Usually, the system design requires the mean deviation to be less than 0.5 mm. Selecting 11,000 training sets is a more appropriate choice. In actual applications, the amount of data can be increased or reduced according to the precision need.

The calibration errors at the working distances of 1.8 m, 1.9 m, 2 m, 2.1 m, and 2.2 m were determined using the same test set (*Nt* = 200) to determine the relationship between the calibration effect and projection distance. Table 2 shows the results of the calibrated mean and maximum deviations. The results show that the calibration effect working distance were not correlated.

Table 3 compares the proposed method with the calibration method for establishing a physical model and the binocular vision calibration method based on the coordinate system. In the calibration method using an established physical model, such as that shown in Figure 3, Equation (5) is used to solve the laser output coordinates (*x*,*y*,*z*). Laser trackers or other laser positioning instruments are used to solve the corresponding coordinates in the world coordinate system. In this method, 12 × 12 reflective points are usually selected to form a lattice, and the conversion relationship is solved using the Newton iterative algorithm, the Levenberg–Marquardt algorithm and the hybrid particle swarm algorithm [9]. The instrument design of this method is relatively complicated. In addition, systematic errors occur owing to the difference between the galvo scanner processing technology and ideal parameters. Calibration time is shorter with this method, but the system needs recalibration when the position of the target to be projected changes. This method is the standard system calibration method.

The binocular vision calibration method based on the coordinate system also establishes the galvo scanner coordinate system and determines the relationship between the scanning galvanometer and camera coordinate systems. This method is also affected by the processing technology of the galvo scanner. The proposed method uses binocular cameras for the calibration and directly determines the conversion relationship between the driving values of the galvanometer and coordinates of the camera coordinate system without establishing the coordinate system of the galvo scanner. In addition, the instrument design in this method is simpler. When the position of the target to be projected changes, the proposed method uses the binocular camera to locate the target, simplifying the process operation.

## 4. Conclusions

This study explored the calibration method for a laser 3D projection system based on binocular vision. First, the positioning principle of binocular vision was analyzed. The basic process of binocular vision positioning was briefly described, and the necessity of the positioning function of binocular vision in the system calibration is explained. Accordingly, two calibration methods for laser 3D projection systems based on binocular vision positioning function were proposed. One method uses binocular vision to simplify the process of solving the mathematical model of the relationship between the deflection values of the galvo scanner and coordinates. The other method is based on the corresponding data of the coordinates of the camera coordinate system and the deflection values of the galvo scanner and uses neural networks to solve the relationship between them. Finally, this study used the second method to perform experiments and analyze the experiment results. The experiment results show that the proposed method improved the calibration accuracy and convenience compared with other traditional methods. Moreover, the proposed method exhibited a relatively good calibration effect.

## Figures and Tables

**Figure 1 sensors-23-01941-f001:**
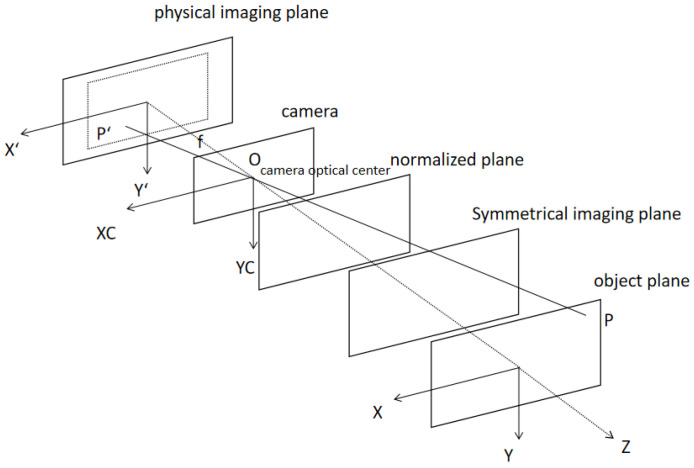
Position diagram of a coordinate system in a camera imaging model.

**Figure 2 sensors-23-01941-f002:**
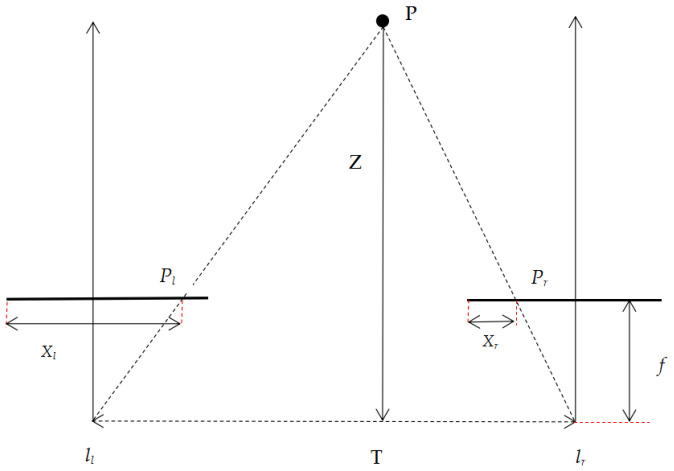
Physical model of binocular vision localization.

**Figure 3 sensors-23-01941-f003:**
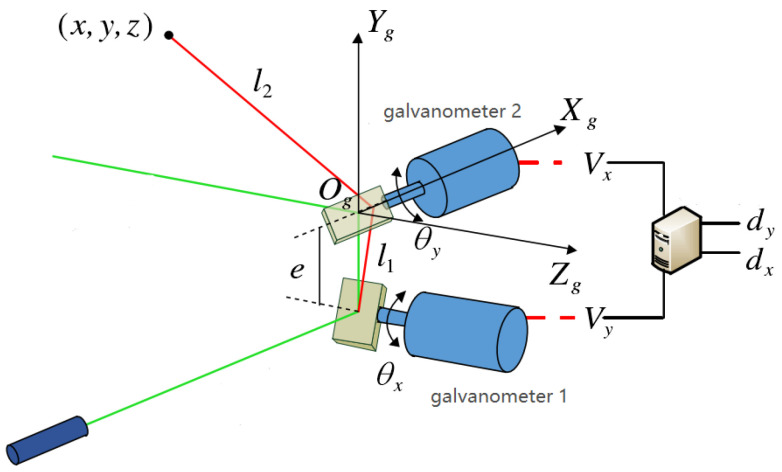
Physical model diagram of 2D galvo scanner. The green lines indicate that the laser is emitted along the coordinate axis, and the red lines indicate that the laser is emitted along any direction.

**Figure 4 sensors-23-01941-f004:**
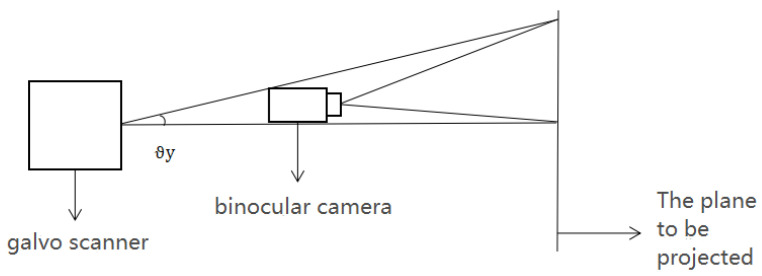
Mapping of camera and galvanometer coordinates.

**Figure 5 sensors-23-01941-f005:**
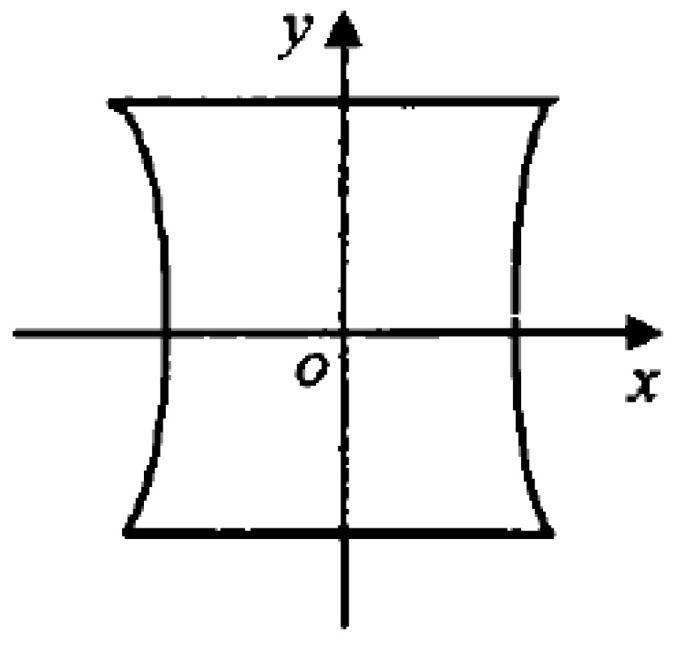
Schematic of nonlinear relationship between galvanometer driving and projection coordinates.

**Figure 6 sensors-23-01941-f006:**
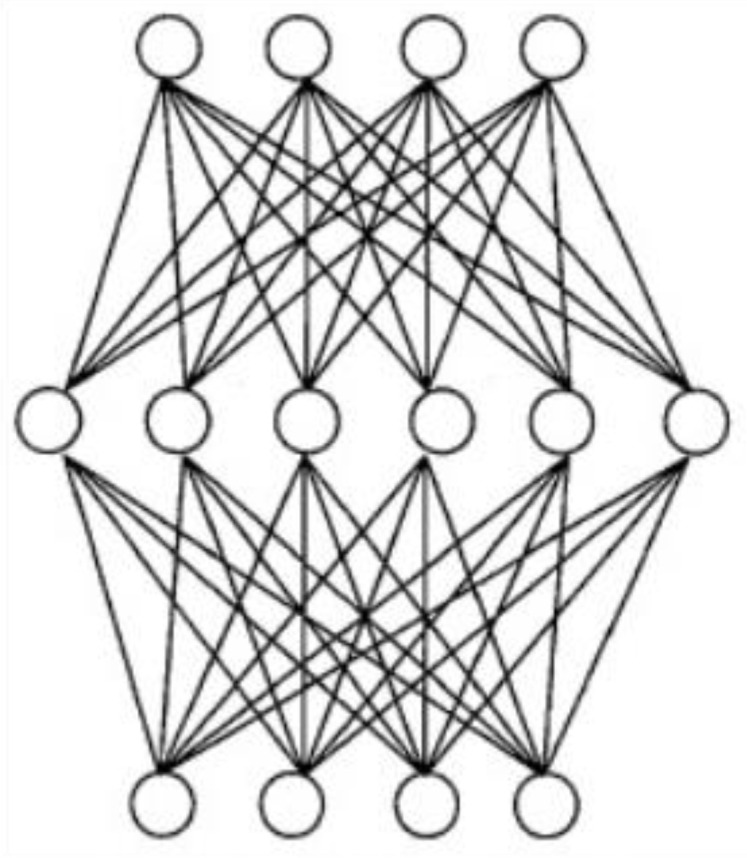
Schematic of a single hidden layer neural network.

**Figure 7 sensors-23-01941-f007:**
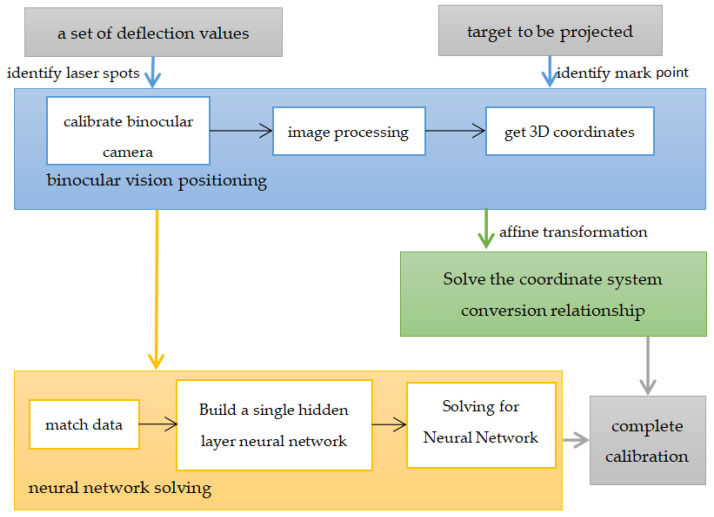
The block-level diagram of the calibration.

**Figure 8 sensors-23-01941-f008:**
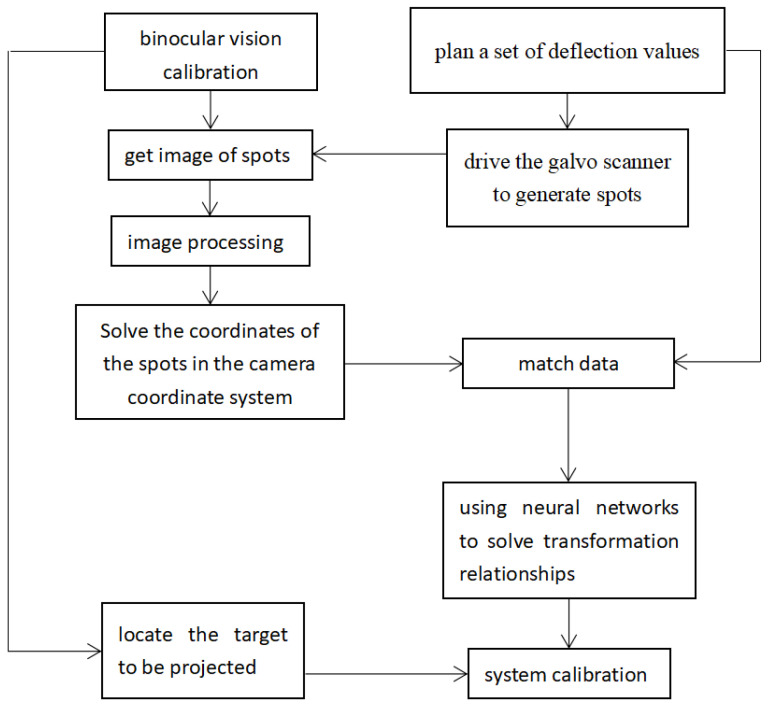
Calibration flow chart based on binocular vision.

**Figure 9 sensors-23-01941-f009:**
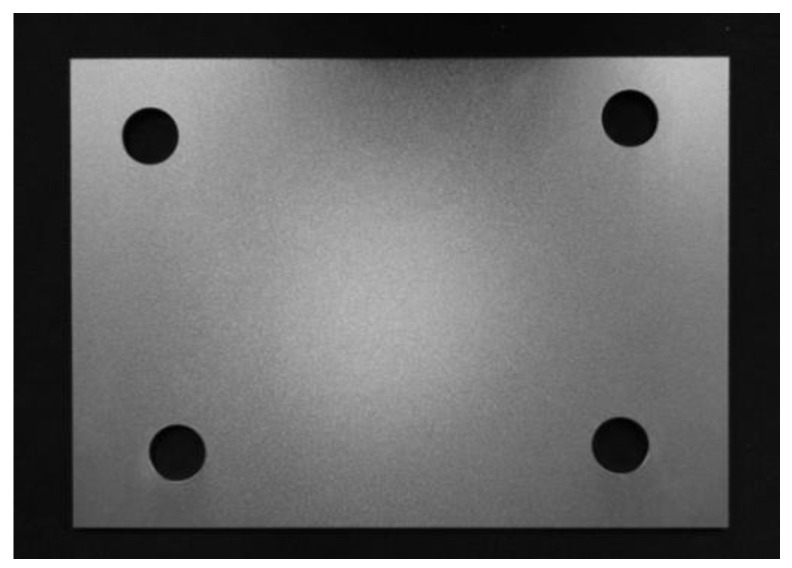
Calibration metal plate.

**Figure 10 sensors-23-01941-f010:**
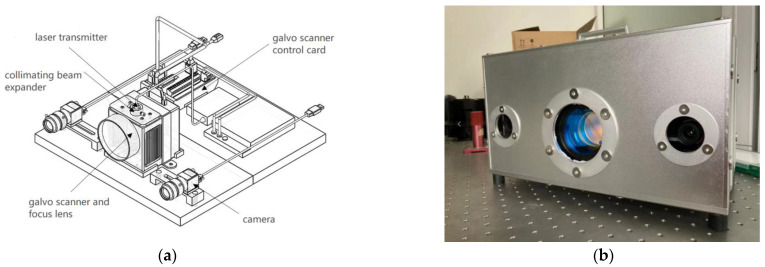
(**a**) System hardware design schematic diagram and (**b**) actual laser 3D projection system.

**Figure 11 sensors-23-01941-f011:**
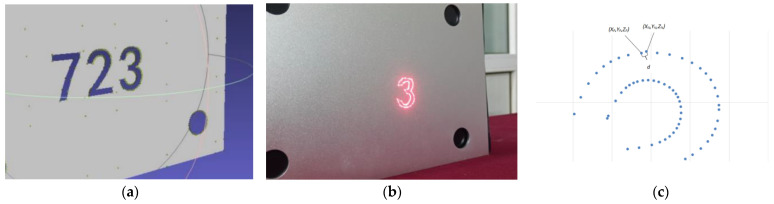
(**a**) The coordinates to be projected on the CAD model, in the shape of the number “723”, (**b**) actual projected coordinates, like the number “3” and (**c**) System Error Diagram.

**Figure 12 sensors-23-01941-f012:**
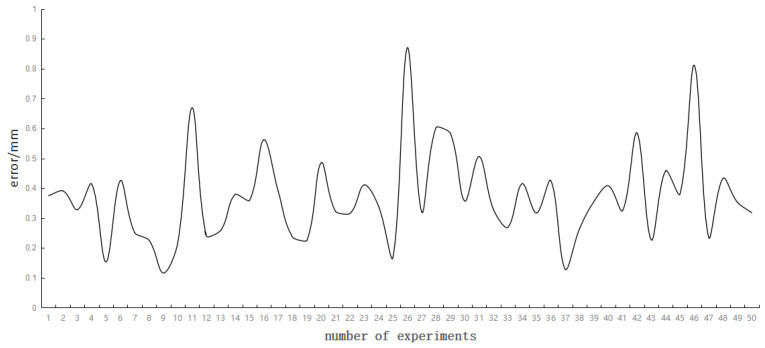
Laser spot error distribution of the calibration result points (average of every 20 groups).

**Figure 13 sensors-23-01941-f013:**
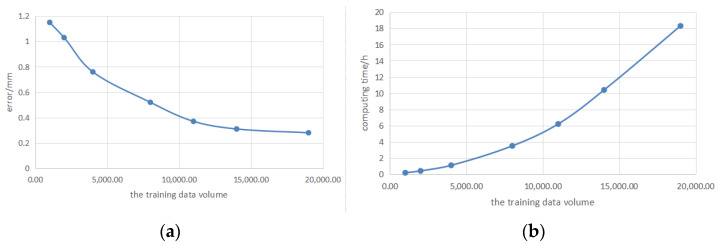
(**a**) The relationship between the training data volume and the average error and (**b**) the relationship between the training data volume and computing time.

**Table 1 sensors-23-01941-t001:** Average and maximum deviations of calibration under different training set data volumes.

Number of Training Sets	Number of Hidden Layer Neurons	Mean Deviation/mm	Maximum Deviation/mm	Computing Time/h
1000	2000	1.15	6.52	0.18
2000	3000	1.03	4.11	0.41
4000	5000	0.76	4.01	1.1
8000	9000	0.52	1.65	3.5
11,000	12,000	0.37	0.87	6.2
14,000	15,000	0.31	0.64	10.4
19,000	20,000	0.28	0.66	18.3

**Table 2 sensors-23-01941-t002:** Average deviation and maximum deviation at different working distances.

Working Distance/m	Mean Deviation/mm	Maximum Deviation/mm
1.8	0.43	0.75
1.9	0.31	1.41
2.0	0.35	0.79
2.1	0.39	0.95
2.2	0.44	0.52

**Table 3 sensors-23-01941-t003:** Comparison of accuracy and convenience of different calibration methods.

Calibration Method		Mean Deviation/mm	Structural Complexity	Calibration Time
Establish physical model	Newton iterative algorithm [9]	0.88	Complex	2 h Recalibration required
	Levenberg–Marquardt algorithm [9]	0.32		
	Hybrid particle swarm algorithm [9]	0.998		
Binocular vision calibration method based on the coordinate system		0.5	Complex	6 h No recalibration required
Method in this article		0.37	Simple	7 h No recalibration required

## Data Availability

Not applicable.

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
