# Peer review of "Calibrating Laser Three-Dimensional Projection Systems Using Binocular Vision"

_sensors, 2023, doi:10.3390/s23041941_

Round 1

Reviewer 1 Report

Could you provide more details on how lens distortion correction was applied in the proposed method? 

A block-level diagram of the calibration setup would interest readers.

What is the error or uncertainty in establishing the relationship between the scanning galvanometer and camera coordinate systems? 

It is reported that a mean deviation of 0.37 mm was obtained with 11000, training sets. What would be the optimum number to increase the accuracy further? Can you also provide a computing time comparison?

Could authors further verify the calibration method with other standard procedures to demonstrate the superiority of the procedure outlined?

What are the recalibration strategies to verify the accuracy of the calibration data?

Fig: 11- Can authors provide error distribution for all different axes? 

Table 3: Can you provide a comparison for a few (2 or 3) more calibration methods (please provide references as well).

Author Response

Dear Reviewer:

Thank you for your review of this article,I will reply to your review item by item.

The lens distortion correction in this article is only used for binocular vision to solve three-dimensional coordinates, and there is no other application in the calibration process.

This article adds the block diagram as Figure 7.

the error in establishing the relationship between the scanning galvanometer and camera coordinate systems is mainly related to the solution process of the neural network,Amendments have been made in the text of the article,this relationship is analyzed in detail starting from line 344 of the article.

Starting from line 344, this article conducts a more detailed analysis of the relationship between the training data volume and the error, and adds a comparison of computing time.

Starting at line 384, this article provides a comparison with the normalization method and modifies Table 3.

The result analysis in this article is similar to the general machine learning analysis method. After solving the neural network, use the pre-planned verification set to substitute into the neural network to obtain the result.Based on this method, this paper makes some modifications.

This article adds Figure 13 as a relationship between the training data volume and error.

Yours siencerely,

Reviewer 2 Report

This study described a calibration method for 3D laser projection system using binocular vision. They proposed two calibration methods for this projection system. However, their manuscript has very few results, which are not effective for presentation. Authors stated that their calibration result was "The maximum and average deviations of the projection laser spot was 0.87 mm and 0.37 mm". However, I could not find their effective evidence. Please show how to evaluate this. Although they show tables 1 and 2, these results weren't show no evidence. Authors should show their evidence to understand their results for readers. I could not also find comparison between their proposed and conventional calibration. If their proposed calibration is better than conventional calibrations, authors need to show them, because that is an evidence of their new. Above-mentioned is my negative comments.

However, they achieved the mean calibration error was 0.38 mm at a working distance of 1.8-2.2 m. I think this is an excellent result. However, there are no evidence. I can't trust it. Please carefully expose all to understand author's achievement.

Author Response

Dear Reviewer:

Thank you for your review of this article.The result analysis in this article is similar to the general machine learning analysis method. After solving the neural network, use the pre-planned verification set to substitute into the neural network to obtain the result.Based on this method, this paper makes some modifications and gives the data of the validation set.

Starting at line 381, this article provides a comparison with the normalization method and modifies Table 3.

Yours siencerely,

Reviewer 3 Report

The paper describes theoretical and experimental research of calibration methods for laser 3D projection systems using binocular vision.

The theoretical and experimental parts of the research are explained in detail. Obtained results are clearly presented and discussed.

It is straightforward work that can be very interesting to the metrology community.

Please delete “et al.” in line 58 - reference [9] refers to L. Guo's PhD dissertation.

Author Response

Dear Reviewer:

Thank you for your review of this article, this article has been deleted “et al.” in line 58 - reference [9] refers to L. Guo's PhD dissertation And the third chapter of the main text has been revised.This article also made some modifications in the experimental method and result verification.

Yours siencerely,

Reviewer 4 Report

Title: Calibrating Laser Three-dimensional Projection Systems Using Binocular Vision

Authors: D. Lao, Y. Wang, F. Wang, and C. Gao

Manuscript ID: sensors-2119225

In this paper, the authors have attempted to calibrate a galvanometric laser projection system using binocular vision. They highlighted that such a system is beneficial for repeated calibration of a target when its position changes repetitively. Unfortunately, the novel contributions of the paper are not brought out clearly in the introduction. Moreover, the authors have not reviewed the literature adequately in the introduction. After a careful literature search (see Refs 1-3 below), it could be noticed that such binocular vision-based calibration of dual-mirror galvanometer laser scanning systems is already available. Furthermore, such works have already been published in Sensors journal (Ref 1 below). Considering the above factors, the reviewer feels that the manuscript reports repetitive work and contains no unique content. Therefore, the paper may not be suitable for publication in the journal.

1.      Tu, J., & Zhang, L. (2018). Effective data-driven calibration for a galvanometric laser scanning system using binocular stereo vision. Sensors, 18(1), 197.

2.      Qi, L., Zhang, Y., Wang, S., Tang, Z., Yang, H., & Zhang, X. (2015). Laser cutting of irregular shape object based on stereo vision laser galvanometric scanning system. Optics and Lasers in Engineering, 68, 180-187.

3.      Tu, J., & Zhang, L. (2018). Rapid on-site recalibration for binocular vision galvanometric laser scanning system. Optics Express, 26(25), 32608-32623.

Author Response

Dear Reviewer:

Thank you for your review of this article,This article does do some duplication in design and methodology, but it differs from the references 1-3 you listed.

Both this article and reference 1 use the neural network method to solve the problem, but the output of the network in reference 1 is the space vectors of the outgoing laser beams, output of the network in this article is the deflection values of the galvo scanner,The two articles are essentially solving different content.In addition, this article further analyzes the positioning principle of binocular vision and the physical model of the galvo scanner, and compares the method in this paper with Newton iterative algorithm, the Levenberg-Marquardt algorithm and the hybrid particle swarm algorithm.This article is more comprehensive in terms of principle elaboration and experimental content.

The system designed in reference 2 is applied to laser processing, and the system designed in this paper is applied to laser 3D projection, which are different application scenarios.method in reference 2 can be concluded as follows: first the 3D model of the object is measured by the stereo cameras; then 3D spatial laser processing trajectory is derived from the acquired 3D model; finally, the 2D laser processing path is determined by reprojecting the spatial path onto the LGS image plane through the transformation given in Eq.This article directly gives the CAD digital model file of the designed projection target, converts it into a point cloud file, and then performs point cloud processing to determine the projection path,Binocular cameras are only used in the positioning process.Reference 2 uses the method of singular value decomposition, and this paper uses the method of neural network.

The differences between this article and reference 3 refer to reference 1.

Based on your review, this article has made some modifications in the experimental methods and validation of the results.

  1. Tu, J., & Zhang, L. (2018). Effective data-driven calibration for a galvanometric laser scanning system using binocular stereo vision. Sensors, 18(1), 197.
  2. Qi, L., Zhang, Y., Wang, S., Tang, Z., Yang, H., & Zhang, X. (2015). Laser cutting of irregular shape object based on stereo vision laser galvanometric scanning system. Optics and Lasers in Engineering, 68, 180-187.
  3. Tu, J., & Zhang, L. (2018). Rapid on-site recalibration for binocular vision galvanometric laser scanning system. Optics Express, 26(25), 32608-32623.

Yours siencerely,

Round 2

Reviewer 1 Report

Thank you authors for your response to my comments, I am happy with it.

Author Response

Dear Reviewer:

Thank you for your review of this article.This paper has some new revisions based on responses from other reviewers and may differ slightly from the original version.

Yours siencerely,

Reviewer 2 Report

This revised manuscript did not improve, unfortunately. Author should reconsider presentation of experimental results. Author should clarify how to obtain their results. I could not believe their results because there are no evidence. 

Author Response

Dear Reviewer:

Thank you for your review of this article.This article has been modified in the results section as per your suggestion.If you have other comments and suggestions, please let me know, I will continue to modify.

Yours siencerely,

Reviewer 4 Report

Comments to the authors:

I have gone through the replies from the authors and the revised manuscript. I am convinced that the paper has some publishing content. Therefore, it can be published. However, before acceptance, the authors must make the following changes to the manuscript.

1. The authors mentioned, "This article does do some duplication in design and methodology, but it differs from the references 1-3 you listed.”

If they accept some duplication, it is their primary responsibility to cite those papers and highlight the difference between their work and the references highlighted by the reviewer. The reviewer was surprised to note that the authors have ignored this fact. Therefore, the authors must cite the following references in the paper and discuss whatever they have replied to me in the introduction of the manuscript to motivate the present work. The reviewer believes that these are the relevant references that must be cited in the paper.

a.      Tu, J., & Zhang, L. (2018). Effective data-driven calibration for a galvanometric laser scanning system using binocular stereo vision. Sensors, 18(1), 197.

b.     Qi, L., Zhang, Y., Wang, S., Tang, Z., Yang, H., & Zhang, X. (2015). Laser cutting of irregular shape object based on stereo vision laser galvanometric scanning system. Optics and Lasers in Engineering, 68, 180-187.

c.      Tu, J., & Zhang, L. (2018). Rapid on-site recalibration for binocular vision galvanometric laser scanning system. Optics Express, 26(25), 32608-32623.

2.      The authors have presented many well-known equations in the manuscript without citing their sources. They should cite the sources from which those equations are being referred. This will help the other users to consult those references while reproducing the work.

3.      Like comment 2, the authors have compared many methods in Table 3. Even though they have provided references to those methods in the main text. Including references to those methods in the Table is a good idea. The reviewer suggests the authors make this minor change. 

Author Response

Dear Reviewer:

Thank you for your review of this article, This article has been revised item by item according to your suggestion. The references you provided are cited in the last paragraph of the Introduction and compared to the methods in this article. Since the contents of references a and c are basically the same, only reference a is cited in this article.The equations cited in this article are provided with references.Table 3 in the text has also been modified according to your suggestion.

Yours siencerely,
